# New Ruthenium-Cyclopentadienyl Complexes Affect Colorectal Cancer Hallmarks Showing High Therapeutic Potential

**DOI:** 10.3390/pharmaceutics15061731

**Published:** 2023-06-14

**Authors:** Ana Rita Brás, Pedro Fernandes, Tiago Moreira, Julia Morales-Sanfrutos, Eduard Sabidó, Alexandra M. M. Antunes, Andreia Valente, Ana Preto

**Affiliations:** 1Centre of Molecular and Environmental Biology (CBMA), Department of Biology, University of Minho, Campus de Gualtar, 4710-057 Braga, Portugal; aritapbras@gmail.com (A.R.B.); pj.10.pedrofernandes@hotmail.com (P.F.); martinsmoreir@gmail.com (T.M.); 2Institute of Science and Innovation for Bio-Sustainability (IB-S), University of Minho, 4710-057 Braga, Portugal; 3Centro de Química Estrutural, Institute of Molecular Sciences and Departamento de Química e Bioquímica, Faculdade de Ciências, Universidade de Lisboa, Campo Grande, 1749-016 Lisboa, Portugal; 4Proteomics Unit, Centre de Regulació Genòmica (CRG), Barcelona Institute of Science and Technology (BIST), Catalonia, 08003 Barcelona, Spain; julia.morales@crg.eu (J.M.-S.); eduard.sabido@crg.cat (E.S.); 5Department of Experimental and Health Sciences, Universitat Pompeu Fabra, 08003 Barcelona, Spain; 6Centro de Química Estrutural (CQE), Institute of Molecular Sciences, Departamento de Engenharia Química, Instituto Superior Técnico (IST), Universidade de Lisboa, 1049-001 Lisboa, Portugal; alexandra.antunes@tecnico.ulisboa.pt

**Keywords:** ruthenium-cyclopentadienyl compounds, colorectal cancer, active targeting, passive targeting

## Abstract

Colorectal cancer (CRC) is among the most deadly cancers worldwide. Current therapeutic strategies have low success rates and several side effects. This relevant clinical problem requires the discovery of new and more effective therapeutic alternatives. Ruthenium drugs have arisen as one of the most promising metallodrugs, due to their high selectivity to cancer cells. In this work we studied, for the first time, the anticancer properties and mechanisms of action of four lead Ru-cyclopentadienyl compounds, namely **PMC79**, **PMC78**, **LCR134** and **LCR220**, in two CRC-derived cell lines (SW480 and RKO). Biological assays were performed on these CRC cell lines to evaluate cellular distribution, colony formation, cell cycle, proliferation, apoptosis, and motility, as well as cytoskeleton and mitochondrial alterations. Our results show that all the compounds displayed high bioactivity and selectivity, as shown by low half-maximal inhibitory concentrations (IC_50_) against CRC cells. We observed that all the Ru compounds have different intracellular distributions. In addition, they inhibit to a high extent the proliferation of CRC cells by decreasing clonogenic ability and inducing cell cycle arrest. **PMC79**, **LCR134**, and **LCR220** also induce apoptosis, increase the levels of reactive oxygen species, lead to mitochondrial dysfunction, induce actin cytoskeleton alterations, and inhibit cellular motility. A proteomic study revealed that these compounds cause modifications in several cellular proteins associated with the phenotypic alterations observed. Overall, we demonstrate that Ru compounds, especially **PMC79** and **LCR220**, display promising anticancer activity in CRC cells with a high potential to be used as new metallodrugs for CRC therapy.

## 1. Introduction

Colorectal cancer (CRC) is a leading cause of morbidity and mortality worldwide. This type of cancer is the third most frequent, as well as the second most common cause of death all over the world [1]. In 2020, more than 1.9 million new cases and 935,000 deaths were estimated worldwide [1]. Taking into account time profiles and population projections, the global burden of CRC is estimated to growth by 60% to over 2.2 million new cases and 1.1 million deaths by 2030 [2]. Conventional therapeutical strategies for CRC consist of surgical resection followed by chemotherapy with 5-fluorouracil (5-FU). 5-FU is the most widely used agent to treat CRC and can be used alone or in combination with other drugs. However, it possesses success rates as low as 10–15%, due to severe side effects and resistance problems [3,4,5].

The limitations of current therapeutic strategies instigate researchers to discover new and more effective therapeutic alternatives. In the last decades, transition metal-based compounds have been widely studied as anticancer drugs [6,7,8]. In this field, platinum-based drugs constitute the most successful group of anticancer agents. Cisplatin, oxaliplatin, and carboplatin are employed in the treatment of various types of cancers, such as CRC [9]. Despite their clinical success, platinum-based drugs have many associated disadvantages and chemotherapy is frequently accompanied by severe side effects and treatment resistance [10].

Among the numerous metal complexes investigated to date, ruthenium compounds have emerged as some of the most promising metallodrugs. Among their other characteristics, Ru-based compounds have shown good anticancer activity, with some Ru complexes being selective for cancer cells. In addition, they present different modes of action, chemical stability, and structural variety, with diverse ligand bonding modes and achievable redox properties [11,12]. Nowadays, there are some Ru(II) and Ru(III) complexes in phase I and II clinical trials for cancer therapy, namely NKP1339 for colorectal carcinoma treatment and, more recently, TLD1433 used with photodynamic therapy for the treatment of nonmuscle invasive bladder cancer [12,13,14].

The group of Ru(II) organometallic compounds has also drawn much attention in the medicinal chemistry field, presenting promising anticancer activity both in vitro and in vivo. The most studied organoruthenium compounds are those with a piano stool configuration based on a Ru(II)-(*η*^6^-C_6_H_6_) core [15,16]. However, another promising group of piano-stool-structured compounds designated as Ru(II)-cyclopentadienyl complexes (RuCp), based on Ru(*η*^5^-C_5_H_5_) derivatives, is also gaining attention due to their recognized anticancer potential [7,17].

Over the last few years, our group has been dedicated to the development of a family of RuCp complexes, in particular, those bearing 2,2′-bipyridine-based ligands, and their functionalization, as potential anticancer drugs for chemotherapy [7,11,18]. The first RuCp complex of this class that showed promising in vitro results was TM34 (Figure 1) [19,20]. Based on the characteristics of this complex, a second generation of RuCp compounds was designed [18]. This new family comprises a low molecular weight parental compound (Figure 2—**PMC79**), to which a biodegradable and biocompatible polymer was added at the bipyridine ligand, leading to a polymer-ruthenium conjugate (Figure 2—**PMC78**) [11,18]. The incorporation of polylactide, recognized by the FDA for drug delivery applications, increased the molecular weight of the complex, which might benefit in vivo from an enhanced permeation and retention effect (passive targeting) [11,21,22]. Envisaging the benefit from active targeting, our group also developed and started to evaluate the biological activity of new derivatives containing biotin (vitamin B7) in the bipyridine ligand [23,24,25]. Biotin is fundamental for our bodies, having multiple biological functions [26]. This particular targeting approach takes advantage of the constant division, intense metabolic activity, and fast growth of cancer cells, which has to be accompanied by a strong uptake of essential vitamins. To this end, the receptors involved in vitamin internalization are frequently overexpressed on the cell surface [27]. In the case of biotin, the sodium-dependent multivitamin transporter is overexpressed in several cancer cell lines [28]. From our library of biotinylated compounds, two stood out, **LCR134** and **LCR220**. Both compounds showed promising in vitro anticancer activity and preliminary in vivo studies in zebrafish and/or nude mice confirmed they are worth exploring (Figure 2) [23,24,25]. The structural difference between the two compounds lies in the presence of a polymer, similar to **PMC78**, which confers on the **LCR220** complex the possibility to benefit from both passive and active targeting.

Although some studies on the anticancer effects of these compounds have already been performed in breast and ovarian cancer models, so far, no study on the anticancer properties of these compounds have been performed on the CRC model. Furthermore, it is important to note that the different genetic backgrounds between breast and ovarian cancer models and the CRC model do not allow conclusions to be drawn about whether these compounds are promising for CRC therapy. However, due to the CRC burden and the need for specific and effective therapies, studying the anticancer properties of these four Ru compounds in CRC-derived cell lines might have additional value in finding new therapeutic avenues for CRC.

With this in mind, in this work the anticancer characteristics of this group of compounds were evaluated in the CRC model. For this purpose, several biological assays, namely cellular distribution, colony formation, cell cycle distribution, proliferation, and apoptosis analysis, as well as reactive oxygen species (ROS) production and mitochondrial alterations assessment, and alterations in the actin cytoskeleton and motility were performed in different CRC cell lines. In addition, alterations of the cellular proteome induced by these compounds were also analyzed in CRC cell lines.

## 2. Materials and Methods

### 2.1. Compounds under Study

The syntheses of the compounds under study were previously reported by us (**PMC79** [24], **PMC78** [29], **LCR134** [24], and **LCR220** [23]).

### 2.2. Cell Lines and Culture Conditions

The colorectal cancer-derived cell lines SW480 and RKO, harboring KRAS and BRAF mutations, respectively, were obtained from the American Type Culture Collection (Manassas, VI, USA). The noncancerous NCM460 cell line derived from normal colon epithelial mucosa was obtained from INCELL’s (San Antonio, TX, USA) [30]. The cells were grown at 37 °C under a humidified atmosphere containing 5% of CO_2_. RKO cell lines were grown in Dulbecco’s Modified Eagle´s Medium High Glucose (Biowest, Nuaillé, France). Both the SW480 and NCM460 cell lines were grown in Roswell Park Memorial Institute 1640 medium with stable glutamine (Biowest, Nuaillé, France). Both mediums were supplemented with 10% fetal bovine serum (*v*/*v*) (Biowest, Nuaillé, France) and 1% penicillin/streptomycin (*v*/*v*) (Biowest, Nuaillé, France). The cells were subcultured once a week when 80% of confluence was reached and, when necessary, seeded in sterile test plates for the assays. The SW480 and RKO cells were seeded at 1 × 10^5^ cells/mL concentration and the NCM460 at 2 × 10^5^ cells/mL, except for some specific assays.

### 2.3. Compounds Dilution and Storage

Compounds **PMC79**, **PMC78**, and **LCR134** were dissolved in 100% of dimethyl sulfoxide (DMSO) and stored at −20 °C. Compound **LCR220** was dissolved in 30% DMSO/70% MQ H_2_O, filtered (40 μm filters), and stored at −80 °C. Aliquots were prepared in sterile conditions and after thawing were discarded. Cisplatin was dissolved in a sterile filtered solution of sodium chloride (NaCl) 0.9% (*w*/*v*) in MQ H_2_O (22 μm filters), stored at −20 °C, and protected from light.

### 2.4. Cell Viability Analysis Using Sulforhodamine B Assay

The cell lines were seeded in 24-well plates and 24 h later the cells were incubated with different concentrations of the compounds for 48 h. Two negative controls were performed, in which the cells were incubated only with (1) complete growth medium and (2) DMSO (maximum of 0.1% DMSO per well (*v*/*v*)). For the compound **LCR220**, an additional negative control was performed with (3) MQ H_2_O (maximum of 0.3% H_2_O per well (*v*/*v*)). After 48 h of treatment, a Sulforhodamine B (SRB) assay was performed as previously described [25]. The results were expressed relative to the negative control (1), which was considered as 100% of cell growth. The half-maximal inhibitory concentration (IC_50_) was estimated using GraphPad Prism 8 software, applying a sigmoidal dose vs response (variable slope) non-linear regression (*n* = 3). To determine the cytotoxic selectivity of the compounds tested, the selectivity index (SI) was calculated according to the following equation [31]:SI = IC_50_ (normal cell line)/IC_50_ (cancer cell line)(1)

### 2.5. Intracellular Distribution Measured Using Inductively Coupled Plasma Mass Spectrometry

Both CRC cells were seeded into 100 mm Petri dishes and incubated with Ru compounds after 24 h. After 48 h, the cells were washed with ice-cold PBS and treated to obtain a cellular pellet. The cytosol, membrane/particulate, cytoskeletal, and nuclear fractions were extracted using a Fraction-PREP (ab288085, BioVision, Waltham, MA, USA) cell fractionation kit according to the manufacturer’s protocol. The Ru (^101^Ru) content in each fraction was measured with a Thermo X-Series Quadrupole Inductively Coupled Plasma Mass Spectrometry (Thermo Fisher Scientific, San Jose, CA, USA) after digestion of the samples and using the same procedure previously described [32].

### 2.6. Colony Formation Assay

SW480 and RKO cells were seeded in 6-well plates, at a concentration of 500 cells/mL and 600 cells/mL, respectively. After 24 h, the cells were incubated with different concentrations of Ru compounds. The negative control cells were treated with DMSO (maximum of 0.1% of DMSO per well (*v*/*v*)) and H_2_O (maximum of 0.3% H_2_O per well (*v*/*v*)). After 48 h, the cells were washed with PBS and the medium was replaced with fresh medium, renewed every 3 days. Eight days after removing the treatments, the cells were stained as previously described [25]. The colonies were counted manually using ImageJ 1.53a software.

### 2.7. Proliferation Assessment Using Carboxyfluorescein Diacetate Succinimidyl Ester Labeling

The CRC cells were labeled with Carboxyfluorescein Diacetate Succinimidyl Ester (CFSE) before seeding [33]. After adhering for 24 h, the cells were treated as described in Section 2.6. The cells were harvested at different time points of 0 h, 24 h, and 48 h after treatment, and the CFSE median fluorescence intensity was analyzed using flow cytometry. The results were evaluated using FlowJo 10.7.2 software.

### 2.8. Cell Cycle Analysis Using Flow Cytometry

The cells were seeded in 6-well plates and after 24 h were incubated with Ru compounds. After 24 h and 48 h, the cells were collected and processed according to the method described in [34]. Cell-cycle phase data analysis and quantification were performed using FlowJo 10.7.2 software.

### 2.9. Detection of DNA Strand Breaks Using Terminal Transferase dUTP Nick End Labeling Assay

The cell lines SW480 and RKO were seeded in 6-well plates. After 24 h, the cells were exposed to all the compounds. The negative control cells were treated as described above. 48 h later both floating and attached cells were collected, fixed, and stained as previously reported by us [34]. The cells were counted manually using ImageJ 1.53a software.

### 2.10. Cell Death Evaluation Using Annexin V/Propidium Iodide Assay

CRC cells were seeded in 6-well plates and 24 h after seeding, treatments were processed as described in Section 2.6. After the incubation period, the cells were collected and stained as previously described [25]. The samples were analyzed using FlowJo 10.7.2 software.

### 2.11. Determination of Intracellular Reactive Oxygen Species Using Dihydroethidium Assay

The cells were seeded in 6-well plates and after 24 h were incubated with **PMC79**, **LCR134**, and **LCR220**. Hydrogen peroxide (H_2_O_2_) (250 μM and 1 mM for the SW480 and RKO cell lines, respectively) was used as the positive control. The negative control cells were treated as described in Section 2.6. After 24 h and 48 h, both floating and attached cells were collected, washed with PBS, centrifuged at 2000 rpm for 5 min, and incubated with 500 nM Dihydroethidium (DHE) (30 min, 37 °C, in the dark) to detect superoxide anion (O^−^_2_). The fluorescence emission of the oxidized DHE was analyzed using flow cytometry. The results were evaluated using FlowJo 10.7.2 software.

### 2.12. Analysis of Mitochondrial Mass and Mitochondrial Membrane Potential Alterations Using Flow Cytometry

The cells were processed as mentioned in Section 2.11. MitoTracker Green^FM^ (Molecular Probes, Eugene, OR, USA) was used to analyze the relative mitochondrial mass and MitoTracker Red CMXRos (Molecular Probes, Eugene, OR, USA) was used simultaneously to monitor the changes in mitochondrial membrane potential (ΔΨm). The cells were stained with 200 nM MitoTracker Green^FM^ and 200 nM MitoTracker Red CMXRos (30 min, 37 °C, in the dark). Sodium acetate (100 mM and 140 mM for the SW480 and RKO cell lines, respectively) was used as the positive control because it causes mitochondrial dysfunction, as previously observed by our group [35]. The fluorescence emission was analyzed using flow cytometry. The values of the mitochondrial mass were expressed as the mean green fluorescence intensity normalized to the correspondent negative control (w/0.1% DMSO). The values of ΔΨm for each time point were expressed as the ratio between the mean red fluorescence intensity and the mean green fluorescence intensity normalized to the correspondent negative control (w/0.1% DMSO). The results were evaluated using FlowJo 10.7.2 software.

### 2.13. Evaluation of Alterations in F-Actin Cytoskeleton Using Phalloidin Staining

The cell lines SW480 and RKO were seeded in 12-well plates with one coverslip per well. 24 h after seeding, the treatments were processed as described in Section 2.6. 48 h later the cells were fixed and stained as previously reported by us [36]. Representative images were obtained using a fluorescence microscope (Olympus motorized BX63F Upright Microscope) at a magnification of 600×.

### 2.14. Western Blot Analysis

Preparation of the total protein extracts, SDS-PAGE, and Western blots were performed as previously described in [37]. The antibodies used were anti-β-Actin (A5441, Sigma, St. Louis, MO, USA) and anti-GAPDH (GTX100118, Gene Tex, Irvine, CA, USA). Quantification of the specific signal was performed using ImageJ 1.53a software.

### 2.15. Cellular Motility Assessment Using Wound Healing Assay

The SW480 and RKO cell lines were seeded at 7 × 10^5^ cells/mL and 5 × 10^5^ cells/mL, respectively, in 6-well plates. After 48 h, a wound was made with a tip, and the cells were incubated with Ru compounds. The wound areas were photographed at 4 h, 8 h, and 12 h. To analyze the data and measure the wound size at each time point, ImageJ 1.53a software was used.

### 2.16. Proteomic Study

The SW480 cells were seeded in 100 mm Petri dishes for 24 h and then were exposed to the Ru compounds. After 48 h, the cells were washed with PBS and lysed using a 6 M urea solution and a cell scrapper. The lysate was transferred to a microtube and passed through a syringe with a 25 G needle to make the solution less viscous. Protein quantification was performed using a DC™ protein assay kit (5000116, Bio-Rad, Hercules, CA, USA).

The samples (10 μg) were reduced with dithiothreitol (30 nmol, 37 °C, 60 min) and alkylated in the dark with iodoacetamide (60 nmol, 25 °C, 30 min). The resulting protein extract was first diluted to 2 M urea with 200 mM ammonium bicarbonate for digestion with endoproteinase LysC (1:10 (*w*:*w*), 37 °C, 6 h, Wako, cat # 129-02541), and then diluted 2-fold with 200 mM ammonium bicarbonate for trypsin digestion (1:10 (*w*:*w*), 37 °C, o/n, Promega cat # V5113). After digestion, the peptide mix was acidified with formic acid and desalted with a MicroSpin C18 column (The Nest Group, Inc., Ipswich, MA, USA) prior to LC-MS/MS analysis.

The samples were then analyzed using an Orbitrap Fusion Lumos mass spectrometer (Thermo Fisher Scientific, San Jose, CA, USA) coupled to an EASY-nLC 1200 (January 2019) (Thermo Fisher Scientific (Proxeon), Odense, Denmark). Peptides were loaded directly onto the analytical column and were separated with reversed-phase chromatography using a 50 cm column with an inner diameter of 75 μm, packed with 2 μm C18 particles spectrometer (Thermo Fisher Scientific, San Jose, CA, USA). The chromatographic gradients started at 95% buffer A and 5% buffer B with a flow rate of 300 nl/min, and gradually increased to 25% buffer B and 75% A in 79 min, and then to 40% buffer B and 60% A in 11 min. After each analysis, the column was washed for 10 min with 100% buffer B. Buffer A was 0.1% formic acid in water and buffer B was 0.1% formic acid in 80% acetonitrile (January 2019).

The mass spectrometer was operated in the positive ionization mode with the nanospray voltage set at 2.4 kV and the source temperature at 305 °C. Acquisition was performed in the data-dependent acquisition mode and with full MS scans, with 1 micro scans at a resolution of 120,000 used over a mass range of m/z 350–1400, with detection in the Orbitrap mass analyzer. The auto gain control (AGC) was set to ‘standard’ and the injection time to ‘auto’. For each cycle of the data-dependent acquisition analysis, following each survey scan the most intense ions above a threshold ion count of 10,000 were selected for fragmentation. The number of selected precursor ions for fragmentation was determined using the “Top Speed” acquisition algorithm and a dynamic exclusion of 60 s. Fragment ion spectra were produced via high-energy collision dissociation at a normalized collision energy of 28% and they were acquired in the ion trap mass analyzer. AGC MS2 was set to 2E4 and an isolation window of 0.7 m/z and a maximum injection time of 12 ms were used. The digested bovine serum albumin (New England biolabs cat # P8108S) was analyzed between each sample to avoid sample carryover and to assure the stability of the instrument. Qcloud was used to control instrument longitudinal performance during the project [38,39]. 

The acquired spectra were analyzed using the Proteome Discoverer software suite (v2.3, Thermo Fisher Scientific) and the Mascot search engine (v2.6, Matrix Science, Singapore) [40]. The data were searched against a Swiss-Prot human database (as of June 2020, 20,406 entries) plus a list of common contaminants and all the corresponding decoy entries [41]. For peptide identification, a precursor ion mass tolerance of 7 ppm was used for MS1 level, trypsin was chosen as the enzyme, and up to three missed cleavages were allowed. The fragment ion mass tolerance was set to 0.5 Da for MS2 spectra. Oxidation of methionine and N-terminal protein acetylation were used as variable modifications, whereas carbamidomethylation on cysteines was set as the fixed modification. The false discovery rate (FDR) for peptide identification was set to a maximum of 5%. Only proteins with at least two unique peptides identified were considered for quantification.

Peptide quantification data were retrieved from the “Precursor ion area detector” node from Proteome Discoverer (v2.3), using a 2 ppm mass tolerance for the peptide-extracted ion current (XIC). The obtained values were used to calculate protein fold changes and their corresponding adjusted *p*-values (*q*-values). The raw proteomics data have been deposited to the PRIDE repository with the dataset identifier PXD041324 [42].

### 2.17. Statistical Analysis

The results were obtained from at least three independent experiments and expressed as mean ± SD. A one-way ANOVA with Dunnett’s post-test or Tukey’s post-test, and a two-way ANOVA with Dunnett’s post-test were used to analyze the results. *p*-values lower than 0.05 were considered statistically significant. All statistical analyses were performed using GraphPad Prism version 8 for macOS, GraphPad Software, San Diego, CA, USA, www.graphpad.com, accessed on 12 May 2023.

## 3. Results

### 3.1. Ru Compounds Decrease Cell Growth of CRC Cells at Low Doses

The effect of **PMC79**, **PMC78**, **LCR134**, and **LCR220** on cell growth was assessed using an SRB assay in the SW480 and RKO CRC-derived cell lines at 48 h of incubation. The results show that all the Ru compounds inhibit cellular growth in a dose-dependent manner. In addition, all the compounds proved to be highly active, presenting a significant decrease in cell growth at low doses (Figure A1). 

Using dose-response curves, the IC_50_ of each compound was calculated and the values are listed in Table 1. All the Ru compounds showed IC_50_ values in the micromolar range. The RKO cells were more sensitive than the SW480, presenting lower IC_50_ values, except for **LCR220**. In addition, all the Ru compounds demonstrated a higher inhibition of cell growth (lower IC_50_ values) than the chemotherapeutic drug cisplatin (Table 1 and Figure A2). Concerning the SW480 cells, only **PMC78** and **LCR220** showed a higher inhibitory effect than cisplatin.

The IC_50_ values of Ru compounds and cisplatin were also calculated in the NCM460 cell line (noncancerous cell line), which allowed the SI to be determined. All the Ru compounds showed higher IC_50_ values in NCM460 than in the CRC cells, except for cisplatin, resulting in a SI higher than one for all the compounds (Table 1). Overall, the CRC cells were more sensitive to the Ru compounds than the noncancerous cell line, which indicates that Ru compounds are selective to CRC cells [43].

### 3.2. Ru Compounds Are Differently Distributed in CRC Cells

The intracellular distribution of the Ru compounds was evaluated in CRC cells. Cytoskeletal, cytosol, membrane, and nucleus fractions were extracted using a commercial kit, as described in the Experimental Section. As shown in Figure 3, **PMC79** and **LCR134** are preferentially distributed in the membrane fraction in both CRC cell lines. On the other hand, **PMC78** is mainly located in the cytoskeleton. Concerning **LCR220**, this compound is found in the cytoskeleton of SW480 cells, while in the RKO cell line, **LCR220** is located in the membrane fraction.

### 3.3. Ru Compounds Reduce the Clonogenic Potential of CRC Cells

A colony formation assay was performed to study the clonogenic ability of the Ru compounds in CRC cells. This assay determines the cell’s capacity to survive exposure to an exogenous agent during a short period of time, and to form colonies after that agent is removed, mimicking in vitro what happens during cycles of chemotherapy. Cisplatin was included as a control in all the studies because it is a metallodrug frequently used in chemotherapy [44].

The SW480 and RKO CRC cell lines were exposed for 48 h to several concentrations of the compounds. One week after the removal of the agents, the colonies were counted.

In the SW480 cells, **PMC79** and **LCR134** significantly decreased colony formation (Figure 4a). On the other hand, **PMC78** and **LCR220** did not affect the clonogenic ability of this cell line. In the RKO cells, all the Ru compounds reduced the ability of cells to form colonies (Figure 4b,c). In addition, **PMC79** and **LCR134** showed the highest ability to decrease the number of colonies formed. Once again, RKO was proved to be the most sensitive cell line.

With respect to cisplatin, this drug reduced the ability of CRC cells to form colonies in both cell lines, in a similar way to **PMC79** and **LCR134** (Figure 4). This is a promising result since cisplatin is one of the most commonly used chemotherapeutic agents in clinics, although it can cause severe side effects and acquisition of resistance [44].

### 3.4. Ru Compounds Inhibit Proliferation in CRC Cells

In order to understand if the Ru compounds affect cell proliferation, CFSE labeling was performed. CFSE is a dye that allows for the monitoring of cell division using flow cytometry. The dye covalently binds to protein amine groups within the cells, resulting in long-lived fluorescent adducts. As viable cells divide, CFSE dye is equally divided between daughter cells, leading to a linear decrease in fluorescence intensity over time.

The SW480 and RKO cells were incubated with increasing concentrations of Ru compounds and cisplatin for 24 h and 48 h (Figure 5 and Figure A3). In the SW480 cells, **LCR134** (14.1 μM and 28.2 μM), **LCR220** (3.6 μM), and cisplatin (7.0 μM and 14.0 μM) decreased cell proliferation capacity at 24 h (Figure 5a and Figure A3a). At 48 h, all the compounds except **PMC78** induced a decrease in proliferation.

With respect to the RKO cells, **LCR220** (5.6 μM) and cisplatin (12.5 μM and 25.0 μM) decreased the proliferation at all time points (Figure 5b and Figure A3b). At 48 h, **PMC79** (3.0 μM and 6.0 μM), **LCR134** (7.7 μM and 15.4 μM), **PMC78** (8.0 μM) induced a decrease in proliferation (Figure 5b).

### 3.5. Ru Compounds Induce Cell Cycle Arrest in CRC Cells

The CFSE results revealed that Ru compounds inhibit proliferation in CRC cells. To understand whether the inhibition of proliferation is a consequence of cell cycle arrest, the effect of the Ru compounds on the cell cycle phase distribution was evaluated using PI staining with flow cytometry. Both CRC cell lines were incubated with Ru compounds for 24 h and 48 h. DNA content histograms revealed two peaks related to the G0/G1 and G2/M phases of the cell cycle (Figure 6 and Figure A4).

In the SW480 cell line, **PMC78** led to an accumulation of cells in the G2/M phase only after 48 h. **LCR134** also led to a cell cycle arrest in the G2/M phase after 24 h and 48 h (Figure 6a and Figure A4a). Regarding **PMC79**, we observed an increase in the percentage of cells in the G0/G1 phase, indicating an arrest in the cell cycle at 24 h, which resulted in an increase in the percentage of cells in the sub-G1 after 48 h of incubation. This alteration in the cell cycle distribution after 48 h of exposure to the **PMC79** compound indicates an increase in the number of dead cells. On the other hand, the substantial increase in the percentage of cells in the G2/M phase that was observed at 24 h after exposure to **LCR220** did not lead to changes in the cell cycle distribution at 48 h. This indicates that the cells were able to recover from the damage caused by the compound at the first time point. Cisplatin treatment also led to alterations in the cell cycle distribution. After 24 h of treatment, an increase in the percentage of cells in the G0/G1 phase was observed. However, after 48 h, cisplatin exposure caused an increase in the number of cells in the S phase and sub-G1.

Concerning the RKO cells, for **PMC79**, **PMC78**, and **LCR134**, a clear accumulation in the G0/G1 phase of the cell cycle, in comparison to the negative control, was observed after 24 h and 48 h (Figure 6b and Figure A4b). Furthermore, an increase in the percentage of cells in the sub-G1 was observed for **LCR220** at 24 h and 48 h. For the cells exposed to cisplatin, an arrest at the G2/M cell cycle phase at 48 h was observed for this drug. In general, the effects of the Ru compounds on cell cycle distribution are different between the Ru compounds and are dependent on the CRC cell line.

### 3.6. Ru Compounds Induce DNA Strand Breaks in CRC Cells

The effect of the Ru compounds on cell death in the CRC cell lines was assessed by performing a Terminal transferase dUTP nick end labeling (TUNEL) assay using an In Situ Cell Death Detection Kit, Fluorescein (11684795910, Roche, Basel, Switzerland).

In comparison to the negative control, all the compounds increased the % of TUNEL-positive cells, although **LCR134** in the SW480 cells was not statistically significant (Figure 7a,b). The Ru compounds **PMC79** and **LCR220** induced the highest levels of DNA strand breaks in SW480 and RKO, respectively (Figure 7c). Apoptotic bodies, phenotypic alterations typical of late apoptosis or necrosis, were observed in both cell lines.

### 3.7. Ru Compounds Induce Apoptosis in CRC Cells

Since DNA strand breaks, present in late apoptotic or necrotic cells, were observed in the CRC cells exposed to Ru compounds, an annexin V/propidium iodide (AV/PI) cytometry-based assay was performed in order to differentiate the type of cell death induced by these compounds. The results show that for the SW480 cells, only cisplatin (7.0 μM and 14.0 μM), **PMC79** (40.0 μM and 80.0 μM), **LCR134** (28.2 μM), and **LCR220** (3.6 μM) compounds induced apoptosis (Figure 8a and Figure A5a). This is more clear when we join the percentage of early apoptotic cells (AV+ PI−) and late apoptotic cells (AV+ PI+), which represent the total number of apoptotic cells (Figure 8c and Figure A5c).

For the RKO cells, only cisplatin (12.5 μM and 25.0 μM), **LCR134** (15.4 μM), and **LCR220** (5.6 μM) compounds significantly induced apoptosis (Figure 8b,d and Figure A5b,d). Importantly, compared to cisplatin, the **PMC79** and **LCR220** compounds induced higher levels of apoptosis in SW480 and RKO, respectively. On the other hand, **PMC78** did not induce apoptosis in any of the cell lines, even at the highest dose.

Relative to the % of necrotic cells, represented by AV- PI+ staining (in orange) (Figure 8c,d and Figure A5c,d), there was a low % of necrotic cells induced by the Ru compounds in both cell lines, which was not significantly different from the negative control. The only exception was for the **PMC79** compound in the SW480 cells, although the % of apoptotic cells was higher than the % of necrotic cells.

### 3.8. Ru Compounds Increase Reactive Oxygen Species Production in CRC Cells

As previously observed, the **PMC79**, **LCR134**, and **LCR220** compounds induced apoptosis in the CRC cells. With this in mind, we sought to determine whether apoptosis might be induced via ROS production. For this purpose, CRC cells exposed to Ru compounds were stained with the ROS probe DHE, to detect O^−^_2_. As a positive control, H_2_O_2_, known to induce ROS, was used.

In SW480, an increase in ROS levels could be observed as early as 24 h, in the conditions where higher levels of apoptosis were observed (**PMC79** 40.0 μM and 80.0 μM, **LCR134** 28.2 μM, and **LCR220** 3.6 μM) (Figure 9). Interestingly, cisplatin, which induced an increase in the levels of apoptotic cells, did not produce a significant increase in ROS levels. 

In the RKO cell line, the levels of ROS were much lower than in SW480, and an increase in the levels of ROS was only observed at 48 h of exposure to **LCR134** (15.4 μM) and cisplatin (12.5 μM and 25.0 μM) (Figure 10). In the case of **LCR220** (5.6 μM), the levels of ROS were statistically higher at 24 h and 48 h. The increase in the ROS levels induced by the Ru compounds was similar to the H_2_O_2_ conditions in both cell lines.

### 3.9. Ru Compounds Increase Mitochondrial Mass and Induce Changes in Mitochondrial Membrane Potential in CRC Cells

Since mitochondria is an organelle involved in apoptotic cell death, the effects of **PMC79**, **LCR134**, and **LCR220** compounds on mitochondrial mass and ΔΨm were evaluated using Mitotracker Green and Mitotracker Red CMXRos. In this assay, sodium acetate was used as a positive control, as it had previously been observed to induce mitochondrial dysfunction in CRC cells [35].

The results demonstrate a trend towards an increase in mitochondrial mass over time in both CRC cell lines (Figure 11 and Figure A6). This effect was consistent in the conditions where an increase in ROS production was observed. With respect to ΔΨm, there was a trend toward an increase in ΔΨm at 24 h, followed by a decrease at 48 h, although this was not significant in all conditions (Figure 12). With respect to the positive control, a decrease in ΔΨm was only observed in the RKO cells.

### 3.10. Ru Compounds Induce Alteration in the Actin Cytoskeleton of CRC Cells

Previous results from this group have revealed that the Ru compounds under investigation affect the actin cytoskeleton of breast cancer cell lines [11]. Moreover, the intracellular distribution study also demonstrated that some Ru compounds are preferentially located in the cytoskeleton fraction. In order to assess the effects of Ru compounds on the actin cytoskeleton structure of CRC cells, the F-actin organization and morphology were studied using phalloidin.

In the SW480 cells, all the compounds, including cisplatin, appeared to affect cell–cell adhesion and intercellular contact establishment, followed by changes in cell phenotype and roundness (Figure 13 and Figure A7). **PMC79**, **LCR134**, **LCR220**, and cisplatin also affected cell number. In addition, **PMC79** seemed to induce filopodia-like protrusions, although these were less evident when compared to RKO.

In the RKO cells, **PMC79** appeared to affect cell cytoskeleton organization with cell dispersion and evident filopodia-like protrusions (Figure 14). Contrary to what was observed in SW480, treatment with **PMC78** did not seem to affect cell–cell adhesion and cell junction establishment. Moreover, **LCR134** and **LCR220** seemed to have no effects on F-actin organization (Figure A8). Cisplatin seemed to influence cell number and cell size without changes in F-actin organization.

Since the Ru compounds induced alterations in the actin cytoskeleton, β-actin expression levels were assessed using Western blot analysis, using the same conditions. No compounds led to changes in β-actin expression levels, except **PMC79** in the SW480 cell line, where a decrease in the expression of this protein was observed (Figure 15 and Figure A9).

### 3.11. Ru Compounds Inhibit Cellular Motility in CRC Cells

The effect of the Ru compounds on cellular motility was assessed by performing a wound-healing assay. In the SW480 cells, a significant decrease in wound closure of 10% and 5.6% was observed for **PMC79** (40.0 μM and 80.0 μM, respectively), while a wound closure of 21% was observed for the negative control condition at 12 h (Figure 16a,b). In the RKO cells, only **LCR220** (5.6 μM) significantly decreased the wound closure (2.7%) compared to the negative control (7.5%) at 12 h (Figure 16c,d). Cisplatin results also show that this drug did not decrease the motility of the CRC cells.

### 3.12. Ru Compounds Lead to Alteration of CRC Cellular Proteome

The analysis of the CRC cells’ proteome after treatment with the Ru compounds showed an alteration of the cellular proteome as a response to **PMC79** and **LCR134** (Figure 17a and Appendix A). Few protein abundance alterations were observed when treating the CRC cells with **PMC78** or **LCR220**, probably due to the limited range of biological changes in the experiment. A gene ontology enrichment analysis of **PMC79** and **LCR134** proteome alterations showed an enrichment of the proteins involved in the biological processes related to mitochondrial function, translation, and cellular response to oxidative stress (Figure 17b), in agreement with our previous observations.

## 4. Discussion

The high incidence and high mortality levels of CRC all over the world make CRC treatment highly relevant to society [1]. Classical chemotherapeutical options are limited to 5-FU and its combination with other drugs [45,46,47]. The use of these therapeutic regimens is frequently associated with low efficacy, several side effects, and resistance to treatments [48]. Platinum-based drugs are classic drugs used in clinics for the treatment of various types of cancer; however, as with other drugs, their use is accompanied by several side effects and resistance problems [10,49].

The lack of specific anticancer agents for CRC therapy with increased efficacy and specificity, and the need to overcome the resistance mechanisms of the agents currently used and reducing their range of side effects, highlights that the development of new target drugs is still a major challenge and a relevant clinical issue for CRC. Over the last few years, researchers have devoted much effort to developing alternative metal-based cancer chemotherapies. Within the group of metallodrugs, Ru compounds have emerged as one of the most promising [12]. At present, some Ru(II) and Ru(III) compounds have entered phase I and II clinical trials for the treatment of lung metastases, CRC, and bladder cancer, which brings confidence to the use and study of Ru complexes for cancer therapy [12,13,14].

Previous results from this group have shown that the Ru compounds studied here possess promising anticancer activity against breast and ovarian cancer models [11,23,24,25]. However, so far, nothing is known about the anticancer properties and mechanism of action of these compounds in a model with such a distinct genetic background as the CRC model. In this work we study, for the first time, the effects of **PMC79**, **PMC78**, **LCR134**, and **LCR220** compounds on CRC biological hallmarks, and explore their mechanism of action and cellular targets.

Our results show that all the Ru compounds have low IC_50_ values, with RKO cells being the most sensitive. In addition, the Ru compounds were more active against both CRC cell lines than cisplatin for almost all the compounds. Previous studies have already demonstrated that these four Ru compounds have high bioactivity in cell lines derived from breast and ovarian cancers (A2780, MCF7, and MDA-MB-231) with an IC_50_ lower than cisplatin [11,23,25]. Comparing the doses, the IC_50_ values calculated for the Ru compounds were in the range of the values previously obtained for other cancer models, and also for the same CRC lines with different Ru compounds [11,23,25,34,37].

Moreover, the CRC cell lines were shown to be more sensitive to our Ru compounds than the noncancerous cell line (NCM460), which had higher IC_50_ values than the CRC cells, with the exception of cisplatin. The SI calculation revealed that Ru compounds selectively affect CRC cells (SI > 1) in contrast to cisplatin (SI < 1). Some authors support that a SI higher than one indicates that the cytotoxicity on cancer cells has surpassed that on healthy non-cancer ones, revealing a selectivity to cancer cells [43,50]. Other authors consider a SI equal to two to be an interesting value, meaning that the compound shows double the cytotoxicity for cancer cells as compared to normal cells [51]. Our results demonstrate that all the Ru compounds were selective for CRC cells, contrary to cisplatin. This is a promising result, since one of the major goals of chemotherapy in clinics is that the dose of a drug eliminates the cancer cells without affecting the normal cells. 

Additionally, these results also highlight previous toxicity studies’ results, which have shown that **PMC79** and **LCR134** are well tolerated in the in vivo zebrafish model [24,52]. Both compounds showed anticancer characteristics, but did not impact normal cell proliferation [52]. Furthermore, the preliminary in vivo assay in mice proved that **LCR134** and **LCR220** compounds were tolerable at the maximum dose tested [23]. Overall, our Ru compounds seem to be highly selective to cancer cells without affecting normal cells.

We observed that all the compounds decreased cell growth in the two CRC cell lines studied. We further investigated if the decrease in cell growth was due to the inhibition of cellular proliferation and/or increased cell death. Our results show that the inhibition of cell growth was primarily due to an inhibition of proliferation observed for all the compounds through a decrease in colony formation, the maintenance of CFSE intensity, and cell cycle arrest, starting at 24 h of exposure to the compounds. Subsequently, for some of the compounds, the inhibition of proliferation resulted in cell death by apoptosis, observed through an increase in the sub-G1 population in the cell cycle, an increase in the TUNEL-positive cells, and in the AV-stained cells after 48 h of incubation with **PMC79**, **LCR134**, and **LCR220**. In general, our data demonstrate that in the first 24 h of exposure to our Ru compounds, an inhibition of proliferation was observed in both CRC cells. In addition, at 48 h the effect on proliferation was maintained for **PMC78**, while **PMC79**, **LCR134**, and **LCR220** led to an increase in apoptotic cell death.

Once again, our data are in agreement with previous results from this group. **PMC79**, **PMC78**, **LCR134**, and **LCR220** have been shown to decrease cellular proliferation in MCF7 and MDA-MB-231 breast cancer-derived cell lines, and induce apoptosis as the main type of cell death [11,23,25]. Moreover, other Ru(II)-cyclopentadienyl compounds have also been demonstrated to decrease cell proliferation and induce apoptosis in CRC-derived cell lines [34,37]. 

Considering the mechanism by which our Ru compounds induce apoptosis, in recent years, some studies have shown that mitochondria play a central role in the mechanism of action of Ru compounds [53,54]. This organelle is responsible for cellular metabolism and apoptotic cell death under certain conditions [55]. Mitochondrial alterations, including the loss of ΔΨm, are essential events that occur during drug-induced apoptosis [56,57]. Some studies have demonstrated that Ru compounds decrease ΔΨm, leading to mitochondrial dysfunction. Subsequently, they activate the mitochondrial apoptosis pathway through the expression of pro-apoptotic members of the B-cell lymphoma-2 family, which helps to release cytochrome c and then triggers the cascade reactions of members of the caspase family to induce apoptosis [58,59]. Moreover, other studies have also supported the idea that the mechanisms of action of some Ru compounds comprise the induction of apoptosis by a ROS-mediated mitochondrial dysfunction pathway via increasing ROS levels [60,61,62]. 

**PMC79** and **PMC78** have previously been shown to induce apoptosis and mitochondrial alterations in the MCF7 breast cancer cell line [11]. In addition, our intracellular distribution results indicated that **PMC79** and **LCR134** in both CRC cells and **LCR220** in the RKO cells are mainly distributed in the membrane fraction. This fraction comprises membrane proteins which include cellular organelles and organelle’s membrane proteins, excluding the nuclear membrane proteins [63].

Taking into account the results of the intracellular distribution, ROS production, mitochondrial mass, and ΔΨm, along with our proteomics results, we might suggest that mitochondria is the target of our compounds. Moreover, our results show a time-dependent increase in ROS levels, which is concordant with the increase in the percentage of apoptotic cells observed for the **PMC79**, **LCR134**, and **LCR220** compounds. This ROS production increase is also in agreement with the increase in mitochondrial mass and the tendency for a decrease in ΔΨm under the same conditions. Considering these data, we can conclude that **PMC79** in the SW480 cells, and **LCR134** and **LCR220** in both CRC cell lines induce apoptosis via ROS production and mitochondrial dysfunction.

Our intracellular distribution results also show that **PMC78** and **LCR220** localize at the cytoskeleton level. Previous results have shown that Ru compounds are mainly distributed in the cytoskeleton of breast cancer cells and lead to alterations in the actin cytoskeleton of these cells [11,23]. Considering all the work and knowledge of this group regarding these compounds, it is worth noting a tendency towards intracellular localization at the cytoskeleton level for polymeric compounds, such as **PMC78** and **LCR220**, whereas non-polymeric compounds, such as **PMC79** and **LCR134**, are mainly distributed at the membrane fraction level.

Actin has an important function in cellular adhesion, migration, polarization, mitosis, and meiosis [64]. F-actin is necessary for the maintenance of several cellular functions, including motility, shape, and polarity [65]. Concerning changes in the actin cytoskeleton, our data are in agreement with previous group results showing that all Ru compounds induce alteration in F-actin organization and morphology [11,23]. In addition, the **PMC79** compound also showed a decrease in expression levels of β-actin in the SW480 cells. Overall, our data suggest that our Ru compounds also target the actin cytoskeleton of CRC cells.

Apart from cellular proliferation and cell death, another important hallmark of cancer that needs to be explored in the discovery of a new anticancer compound is cell migration. Cell migration is a fundamental process involved in various biological processes, with a high importance in cancer progression. Our data reveal that **PMC79** and **LCR220** in SW480 and RKO cells, respectively, significantly inhibited migration in CRC cells. Overall, considering the effects of our Ru compounds on the actin cytoskeleton and wound healing assay, we can suggest that our Ru compounds have little effect on cellular migration.

Concerning the different chemical structures of **PMC79**, **PMC78**, **LCR134**, and **LCR220** compounds and their anticancer properties, we expected that the biotinylated compounds could increase the targeting of cancer cells more than the non-biotinylated ones. As we mentioned before, several studies have reported that CRC is one of the cancer models that overexpress sodium-dependent multivitamin transporters in the cell surface [28]. Thus, our hypothesis was that CRC therapy could benefit from the introduction of biotin to the Ru compounds. In general, the biotinylated compounds **LCR134** and **LCR220** showed good anticancer characteristics in terms of selectivity towards cancer cells, inhibition of proliferation, and induction of apoptosis in both cell lines. **LCR220** was demonstrated to have a higher effect in RKO cells, being the only one that inhibited cellular motility and also induced higher levels of apoptotic cell death. Overall, in RKO cells (with a BRAF mutation), **LCR220** was shown to be the most promising compound, suggesting that the presence of biotin might increase the targeting of CRC cells with a BRAF mutation. This mutation is one of the most frequently found in CRC, and for which no specific treatments are available [66].

In summary, here we show, for the first time, that **PMC79**, **PMC78**, **LCR134**, and **LCR220** have promising anticancer properties for CRC treatment. Our data reveal that all Ru compounds are highly active and selective for CRC cells. In addition, all the compounds inhibited proliferation in both CRC cell lines. **PMC79**, **LCR134**, and **LCR220** also induced cell death by apoptosis. **PMC79** stood out in the SW480 cell line, harboring a KRAS mutation, by inhibiting proliferation and migration and leading to a very pronounced increase in apoptosis. In the RKO cell line, with a mutation in BRAF, the compound **LCR220** was the most promising. Similar to **PMC79** in the SW480 cell line, **LCR220** also inhibited proliferation and migration, in addition to inducing apoptosis in RKO cells. For these compounds, the different genetic background of both CRC cell lines seems to have influenced the mechanism of action, which may be an advantage for treating CRC with KRAS or BRAF mutations. The functionalization of **LCR220** with a biopolymer and biotin endowed this compound with anticancer characteristics that make it a promising compound for the treatment of CRCs with a BRAF mutation. Moreover, Ru compounds seem to have the mitochondria and the cytoskeleton as targets, and induce apoptosis via ROS production and mitochondrial dysfunction. Altogether, these results highlight the specific anticancer potential of these new Ru compounds toward CRC cells, offering opportunities for their therapeutic application.

## Figures and Tables

**Figure 1 pharmaceutics-15-01731-f001:**
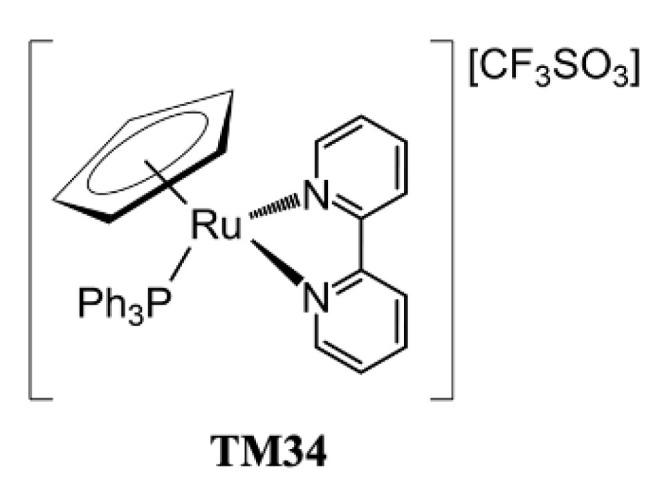
Chemical structure of TM34.

**Figure 2 pharmaceutics-15-01731-f002:**
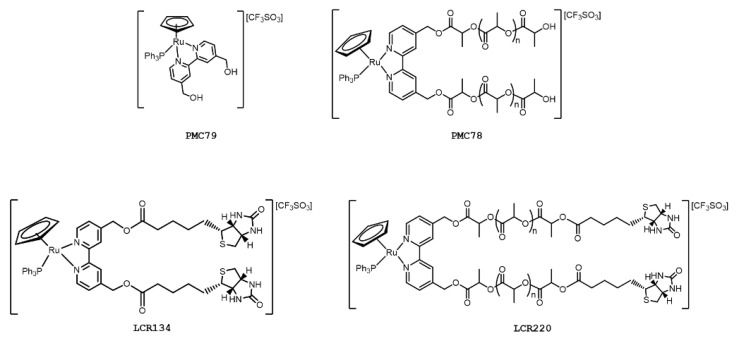
Chemical structures of RuCp compounds **PMC79**, **PMC78**, **LCR134**, and **LCR220**.

**Figure 3 pharmaceutics-15-01731-f003:**
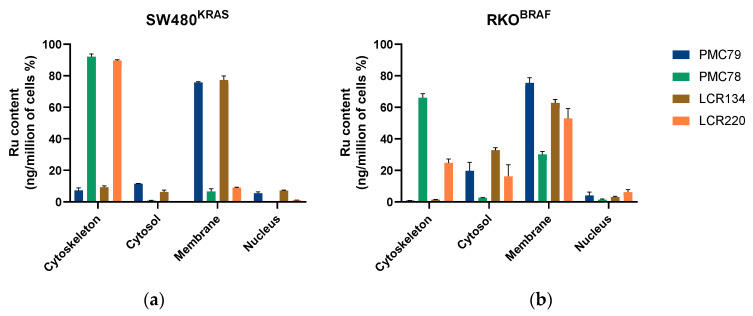
Cellular distribution of Ru compounds in SW480 (**a**) and RKO (**b**) cell lines. Data are presented as mean ± SD of two independent experiments.

**Figure 4 pharmaceutics-15-01731-f004:**
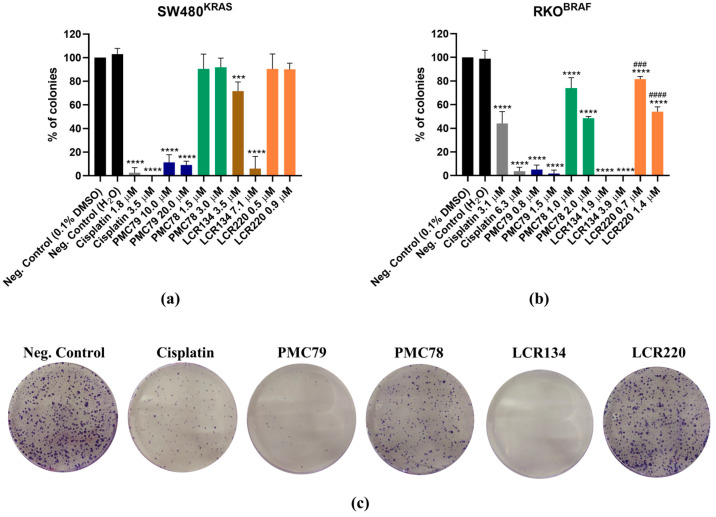
Clonogenic ability of SW480 (**a**) and RKO (**b**) cell lines treated with Ru compounds. Representative images of colony formation in RKO (**c**) cells after incubation with the lowest concentration of Ru compounds. Data are presented as mean ± SD from at least three independent experiments. *** *p* ≤ 0.001 and **** *p* ≤ 0.0001 compared to negative control (0.1% DMSO). ### *p* ≤ 0.001 and #### *p* ≤ 0.0001 compared to negative control (H_2_O).

**Figure 5 pharmaceutics-15-01731-f005:**
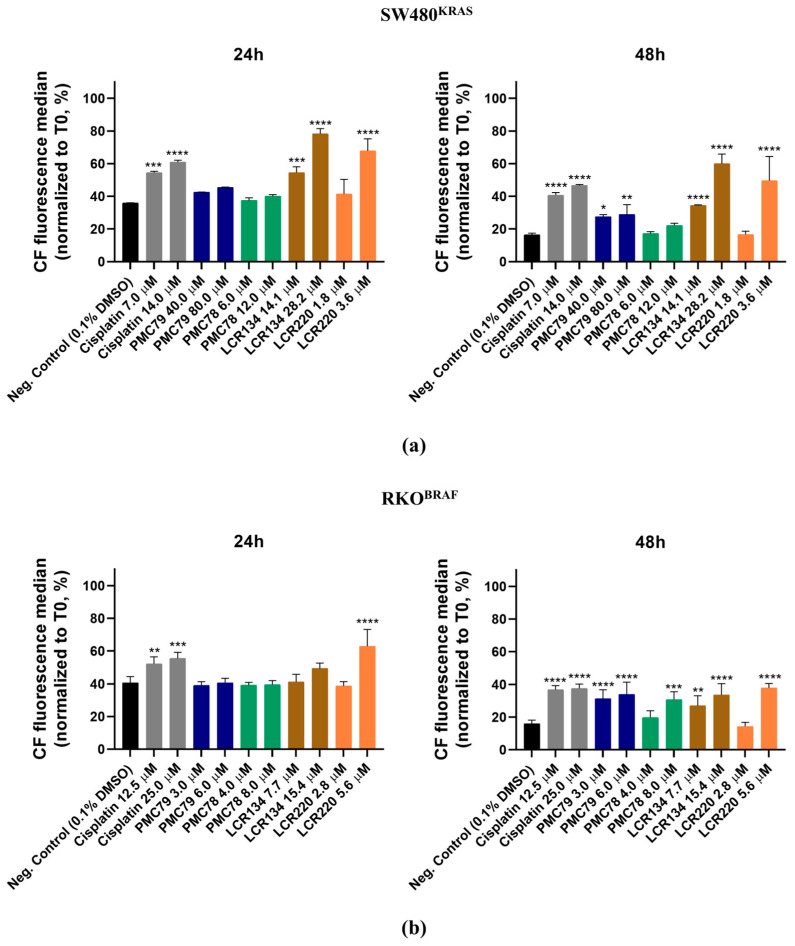
Assessment of Ru compounds proliferation effect in SW480 (**a**) and RKO (**b**) cell lines after 24 h and 48 h of incubation with Ru compounds. Data are presented as mean ± SD from at least three independent experiments. * *p* ≤ 0.05, ** *p* ≤ 0.01, *** *p* ≤ 0.001, and **** *p* ≤ 0.0001 compared to negative control (0.1% DMSO).

**Figure 6 pharmaceutics-15-01731-f006:**
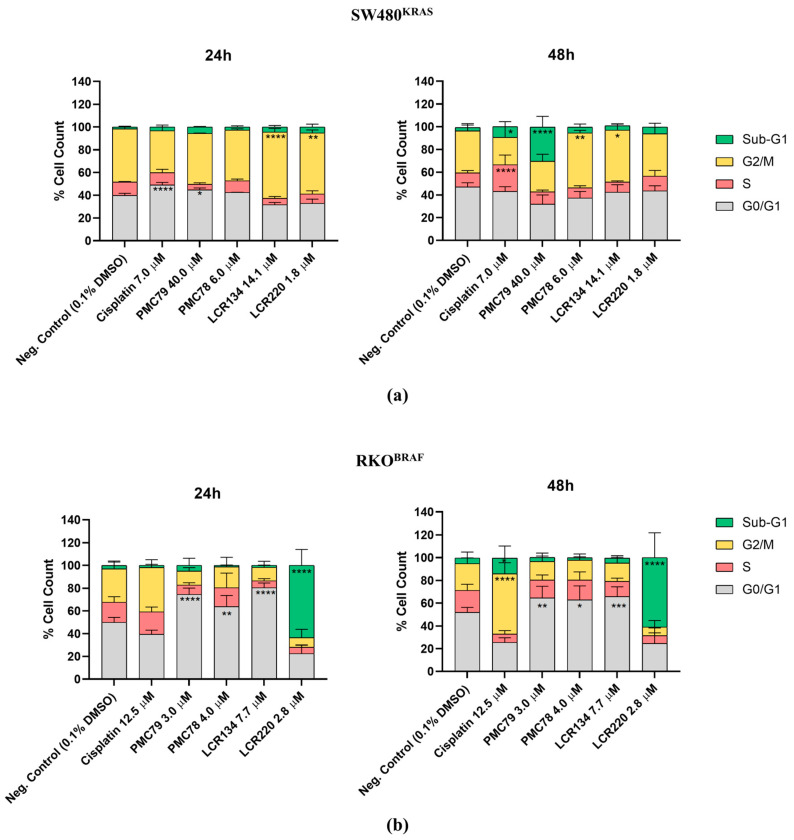
Cell-cycle phases distribution analysis using flow cytometry, after 24 h and 48 h of incubation with Ru compounds, in SW480 (**a**) and RKO (**b**) cell lines. Data are presented as mean ± SD from at least three independent experiments. * *p* ≤ 0.05, ** *p* ≤ 0.01, *** *p* ≤ 0.001, and **** *p* ≤ 0.0001 compared to negative control (0.1% DMSO).

**Figure 7 pharmaceutics-15-01731-f007:**
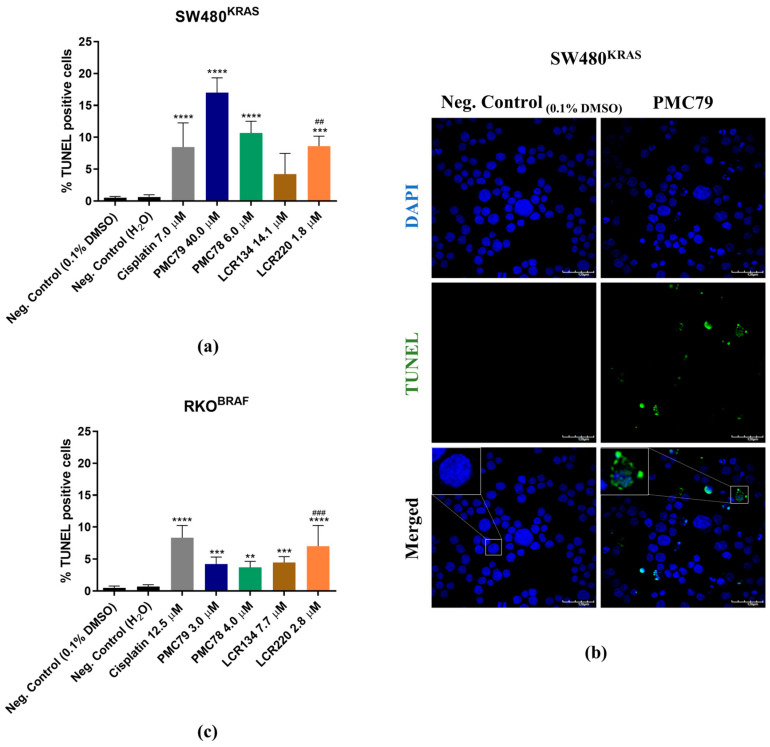
Assessment of DNA strand breaks induced by Ru compounds in SW480 (**a**) and RKO (**c**) cells using a TUNEL assay. Representative images of SW480 cells (**b**) (×600). DAPI (40,6-diamidino-2-phenylindole), FITC (fluorescein isothiocyanate), and their merger were obtained with fluorescence microscopy. Values represent mean ± SD of at least three independent experiments. ** *p* ≤ 0.01, *** *p* ≤ 0.001, and **** *p* ≤ 0.0001 compared to negative control (0.1% DMSO). ## *p* ≤ 0.01 and ### *p* ≤ 0.001 compared to negative control (H_2_O). Scale bar is 120 μm.

**Figure 8 pharmaceutics-15-01731-f008:**
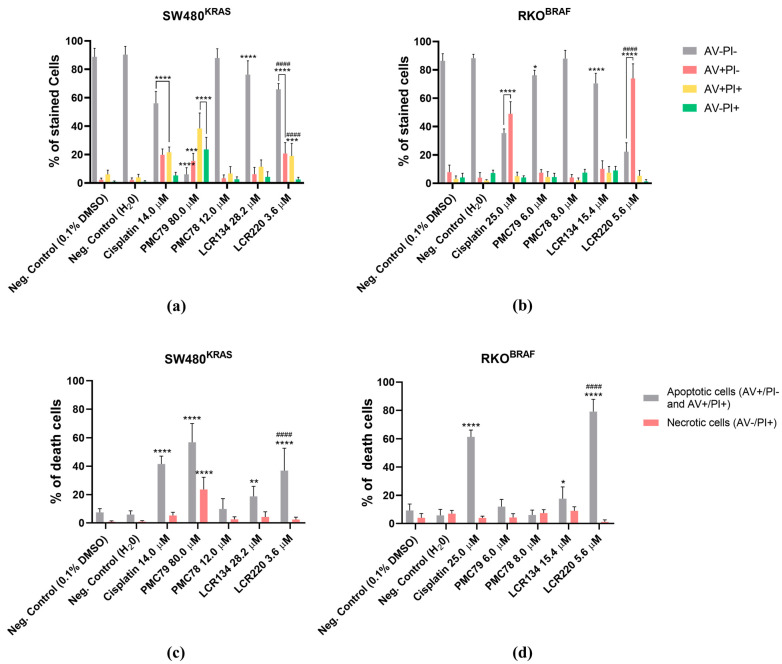
Evaluation of the type of cell death induced by higher concentrations of Ru compounds in SW480 (**a**) and RKO (**b**) cells. Analysis of % of apoptotic cells (AV+PI− and AV+PI+) vs necrotic cells (AV-PI+) in SW480 (**c**) and RKO (**d**) cells. Values represent mean ± SD of at least three independent experiments. * *p* ≤ 0.05, ** *p* ≤ 0.01, *** *p* ≤ 0.001, and **** *p* ≤ 0.0001 compared to negative control (0.1% DMSO). #### *p* ≤ 0.0001 compared to negative control (H_2_O).

**Figure 9 pharmaceutics-15-01731-f009:**
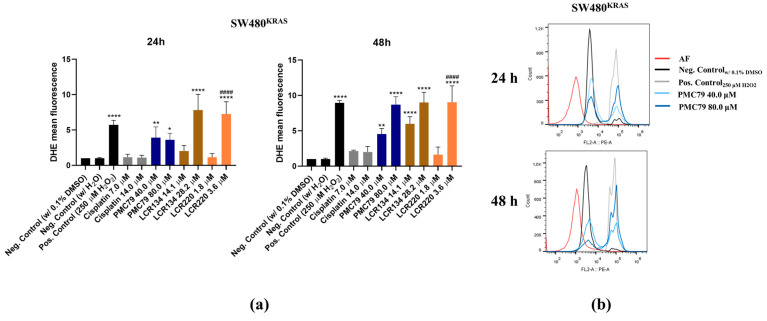
Evaluation of ROS production induced by Ru compounds in SW480 cells (**a**). Representative histograms of **PMC79** in SW480 (**b**). Values represent mean ± SD of at least three independent experiments. * *p* ≤ 0.05, ** *p* ≤ 0.01, and **** *p* ≤ 0.0001 compared to negative control (0.1% DMSO). #### *p* ≤ 0.0001 compared to negative control (H_2_O).

**Figure 10 pharmaceutics-15-01731-f010:**
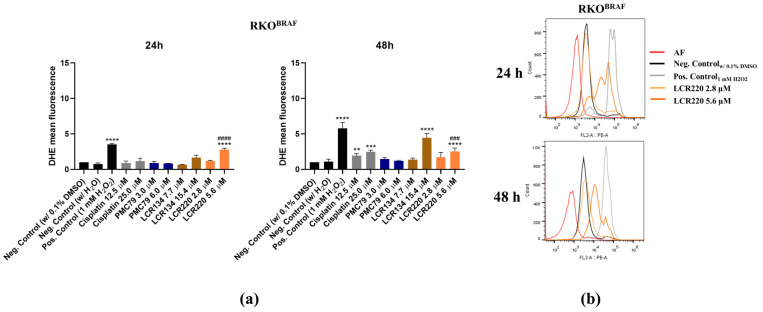
Evaluation of ROS production induced by Ru compounds in RKO cells (**a**). Representative histograms of **LCR220** in RKO (**b**). Values represent mean ± SD of at least three independent experiments. ** *p* ≤ 0.01, *** *p* ≤ 0.001, and **** *p* ≤ 0.0001 compared to negative control (0.1% DMSO). ### *p* ≤ 0.001 and #### *p* ≤ 0.0001 compared to negative control (H_2_O).

**Figure 11 pharmaceutics-15-01731-f011:**
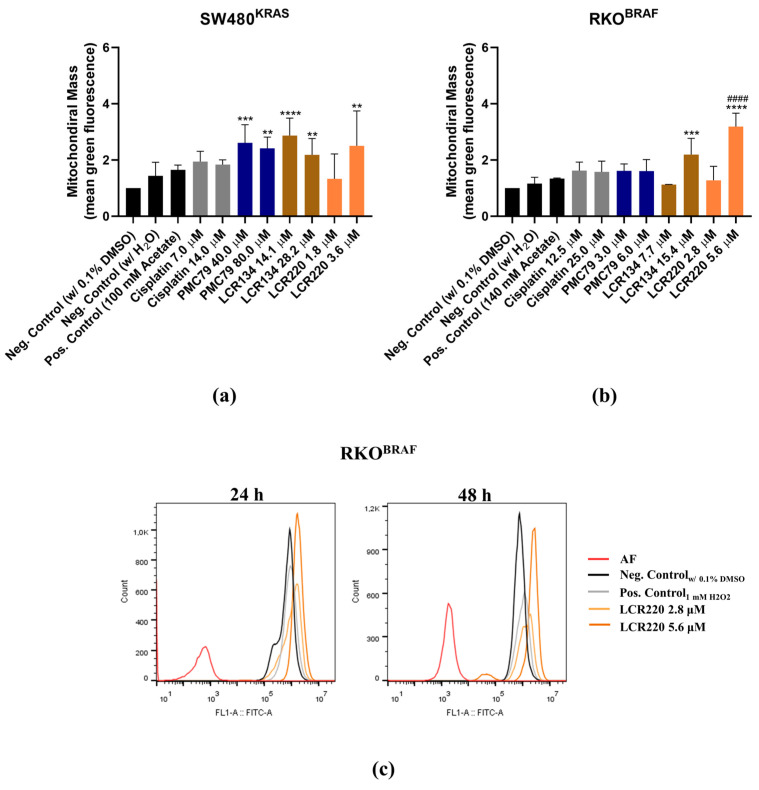
Ru compounds increase mitochondrial mass in SW480 (**a**) and RKO (**b**) cells at 48 h. Representative histograms of **LCR220** in RKO (**c**). Values represent mean ± SD of at least three independent experiments. ** *p* ≤ 0.01, *** *p* ≤ 0.001, and **** *p* ≤ 0.0001 compared to negative control (0.1% DMSO). #### *p* ≤ 0.0001 compared to negative control (H_2_O).

**Figure 12 pharmaceutics-15-01731-f012:**
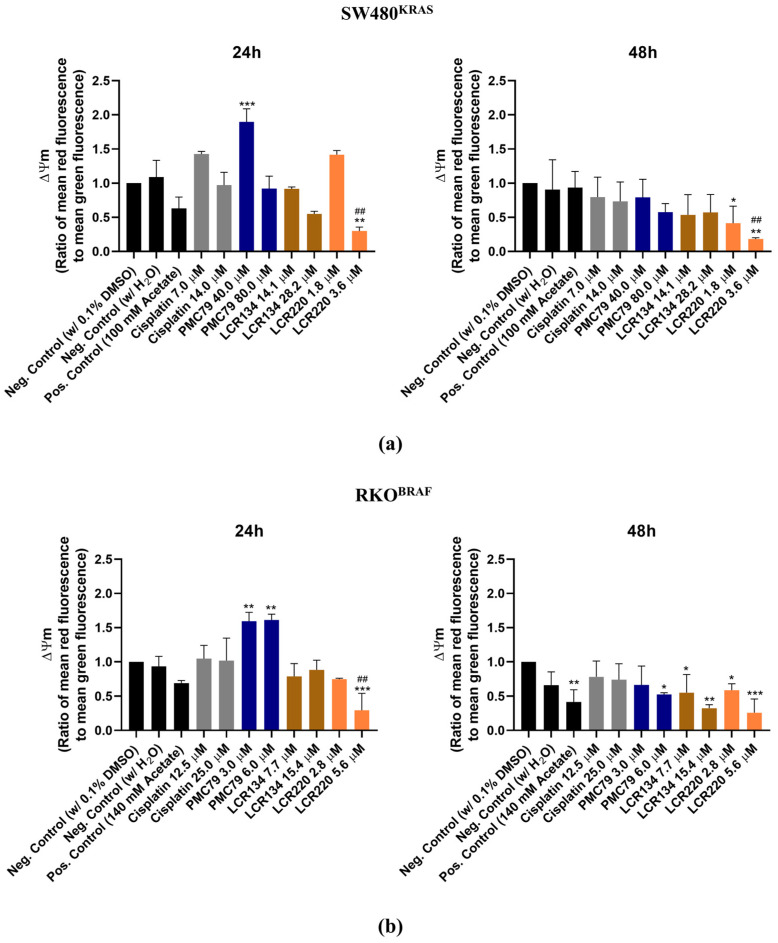
Ru compounds induce alterations in mitochondrial membrane potential in SW480 (**a**) and RKO (**b**) cells. Values represent mean ± SD of at least three independent experiments. * *p* ≤ 0.05, ** *p* ≤ 0.01, and *** *p* ≤ 0.001 compared to negative control (0.1% DMSO). ## *p* ≤ 0.01 compared to negative control (H_2_O).

**Figure 13 pharmaceutics-15-01731-f013:**
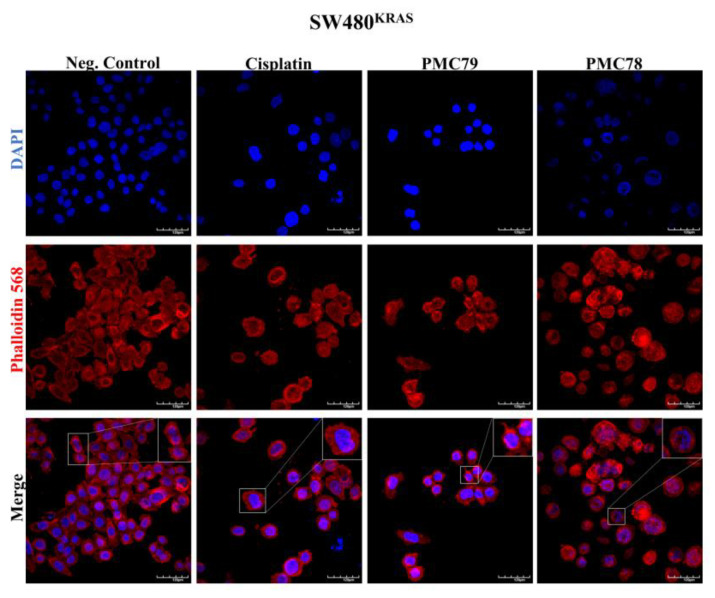
Ru compounds affect the actin cytoskeleton of SW480 cells. Representative images (×600) of DAPI (4′,6diamidino-2-phenylindole), Phalloidin-AlexaFluor^®^ 568, and their merger were obtained with confocal microscopy. The results were obtained from at least three independent experiments. Scale bar for images is 120 μm.

**Figure 14 pharmaceutics-15-01731-f014:**
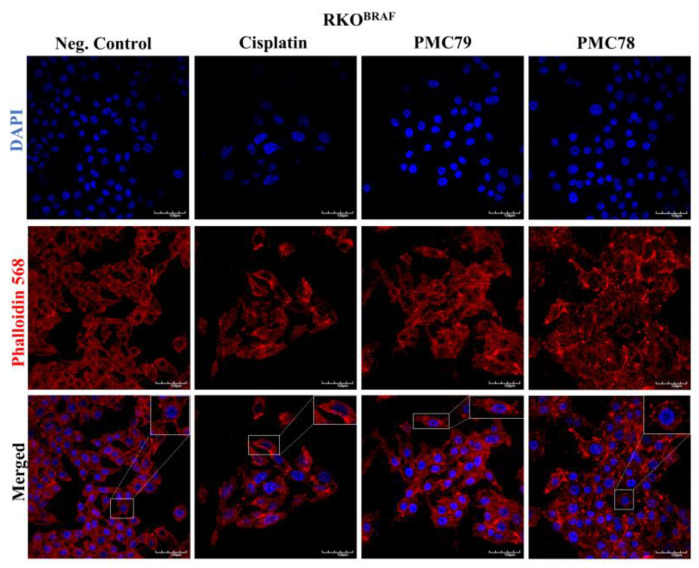
Ru compounds affect the actin cytoskeleton of RKO cells. Representative images (×600) of DAPI (4′,6diamidino-2-phenylindole), Phalloidin-AlexaFluor^®^ 568, and their merger were obtained with confocal microscopy. The results were obtained from at least three independent experiments. Scale bar for images is 120 μm.

**Figure 15 pharmaceutics-15-01731-f015:**
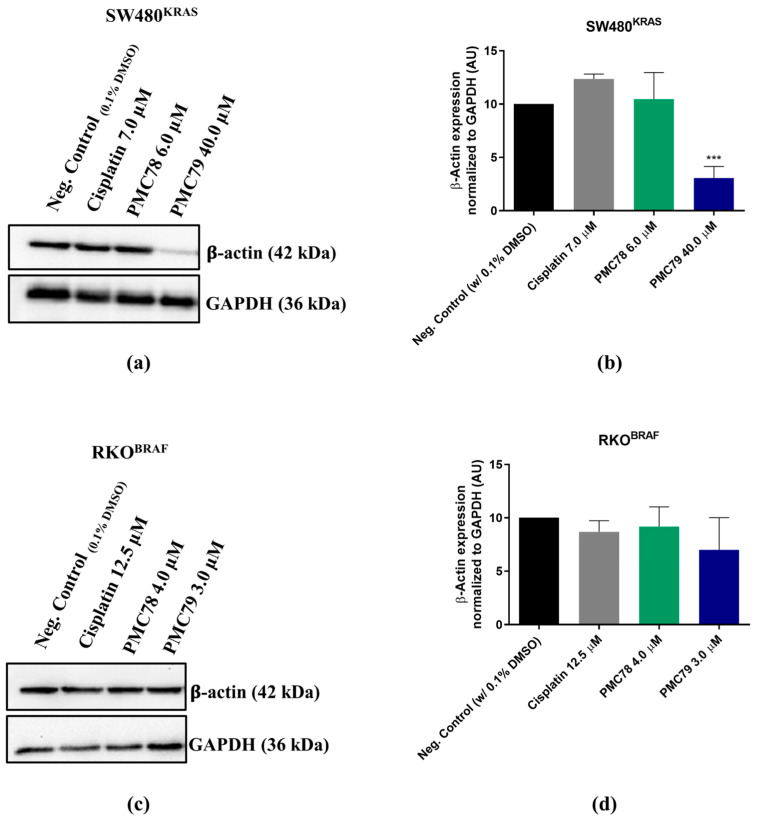
**PMC79** decreases the levels of β-actin in SW480. Representative blots of β-actin expression for SW480 (**a**) and RKO (**c**), and analysis of quantification (**b**,**d**). Data are presented as mean ± SD from at least three independent experiments. *** *p* ≤ 0.001 compared to negative control (0.1% DMSO).

**Figure 16 pharmaceutics-15-01731-f016:**
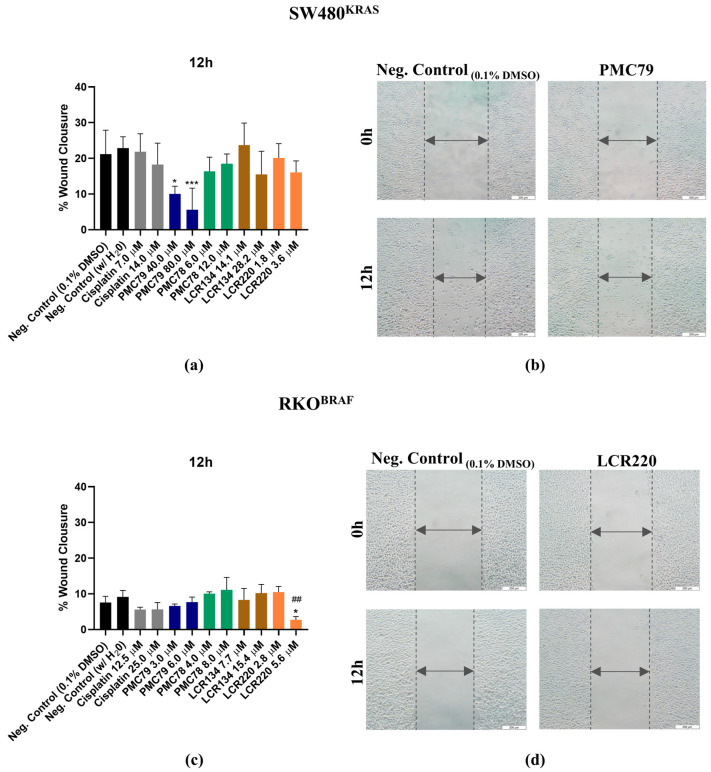
Evaluation of Ru compounds effect on cellular motility in SW480 (**a**) and RKO (**c**) cell lines after 12 h of incubation with Ru compounds. Representative images (×100) of the highest dose of **PMC79** in SW480 (**b**) and **LCR220** in RKO (**d**) cell lines. Data are presented as mean ± SD from at least three independent experiments. * *p* ≤ 0.05 and *** *p* ≤ 0.001 compared to negative control (0.1% DMSO). ## *p* ≤ 0.01 compared to negative control (H_2_O). Scale bar for all images is 200 μm.

**Figure 17 pharmaceutics-15-01731-f017:**
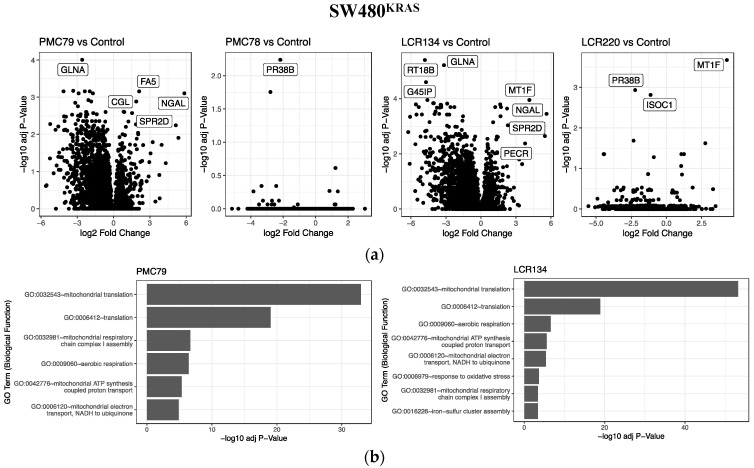
Analysis of cellular proteome alterations induced by Ru compounds in SW480 CRC cell line. (**a**) Protein abundance alterations in SW480 cells treated with **PMC79**, **PMC78**, **LCR134**, and **LCR220**. (**b**) Gene ontology enrichment analysis of **PMC79** and **LCR134** compounds in SW480 cells.

**Table 1 pharmaceutics-15-01731-t001:** Cellular viability of Ru complexes against SW480 and RKO colorectal cancer-derived cell lines and noncancerous cell line NCM460. IC_50_ values were determined at 48 h of incubation.

	IC_50_ (μM)			SI	
**Compounds**	**SW480^KRAS^**	**RKO^BRAF^**	**NCM460**	**SW480**	**RKO**
**PMC79**	40.0 ± 2.0	3.0 ± 0.5	44.0 ± 6.9	1.1	14.7
**PMC78**	6.0 ± 0.2	4.0 ± 0.2	14.3 ± 0.6	2.4	3.6
**LCR134**	14.1 ± 0.7	7.7 ± 0.4	57.2 ± 6.0	4.1	7.4
**LCR220**	1.8 ± 0.1	2.8 ± 0.1	4.3 ± 0.1	2.4	1.5
**Cisplatin**	7.0 ± 0.1 [33]	12.5 ± 1.2 [33]	6.3 ± 0.9	0.9	0.5

Values expressed as mean ± SD.

## Data Availability

Data will be made available on request.

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
