# Peer review of "New Ruthenium-Cyclopentadienyl Complexes Affect Colorectal Cancer Hallmarks Showing High Therapeutic Potential"

_pharmaceutics, 2023, doi:10.3390/pharmaceutics15061731_

Round 1

Reviewer 1 Report

The investigators examined the anti-cancer properties of four synthetic Ru-cyclopentadienyl compounds (PMC79, PMC78, LCR134, and LCR220) on two colorectal cancer (CRC) cell lines: SW480 and RKO, comparing these with normal colon RCM460 cells. Several parameters were evaluated after cell exposure to the above compounds, including Ru uptake, intracellular distribution, cell viability, clonogenicity, proliferation ability, cell motility, cell cycle status, DNA strand breaks, ROS formation, mitochondrial mass, apoptosis vs. necrosis, F-actin /cytoskeletal status, and protein status via proteomic analysis. This is the first study to test the above four Ru compounds on CRC cells, previous work focusing on breast and ovarian cancer cells. Although the objective of this study is is highly important vis-a-bis anti-CRC chemotherapy, there are several problems which detract from its suitability for publication. These are listed as follows:

1. A massive amount of data are presented for the four anti-CMC compounds and two CMC lines, much of which is hard to distinguish in formats presented, e.g. Figs. 4 and 8-11. A clearer way of presenting these data needs to be used, e.g. tables where variables like concentrations are clearly indicated. 

2. Application of two relatively new Ru compounds, LDR134 and LCR220, is described, both of which are biotin-labeled, which is said to make them better targeted to CRC cells than PMC78 and PMC79. If so, this could be a highlight of this study. Although there is some indication of this in Table 1 (IC50 values), it's not convincing overall (e.g. Fig. 7a-c, Fig. 9a-d. In Fig. 14 gap-closure/wound-healing data, there's no better effect of LCR134/220 on SW480 cells than with the other Ru compounds. So attaching biotin doesn't seem to make any difference. 

3. LCR134/220 seem to increase mitochondrial mass more than the other agents, but what does this mean vis-a-vis cytotoxicity? No explanation is offered.

4. Finally, very abundant overall data are presented suggesting that the developed Ru compounds are effective against CRC, but there is little indication that the biotin-linked ones are any better than the non-biotin ones, which fails to support a major premise. In addition, it's not clear that these Ru compounds will have negative effects on other non-cancer cells besides normal colon cells, e.g. vascular endothelial cells. 

Excellent use of English language throughout the manuscript.

Reviewer 2 Report

The work by Ana Rita Brás and co-workers reports anticancer properties and mechanisms of four Ru-cyclopentadienyl compounds, biological assays were performed to evaluate cellular distribution, colony formation, cell cycle, proliferation, apoptosis, motility as well as cytoskeleton and mitochondrial alterations. The results showed that all compounds display high bioactivity and selectivity and promising anticancer activity.

The authors found that the developed Ru compounds are effective against CRC, but there is no data that the biotin-linked ones are better than the non-biotin ones, so the authors should add the data.  

It is a new attempt to try ruthenium compounds as potential anticancer drugs. However, major revision are recommended. 

Minor editing of English language required.

Reviewer 3 Report

The manuscript "New ruthenium-cyclopentadienyl complexes affect colorectal cancer hallmarks showing high therapeutic potential" presents the therapeutic application of Ru-complex-based anticancer drugs using various forms of chelating ligands. I appreciate the author provides enough information for readers. However, some points need to be clear before publication.

1.      The language is ambiguous. There appear very long statements in the abstract and some other parts of the manuscript. Need major revision.

2.      Did the author was informed with the following article C Teixeira-Guedes · 2022. Please explain your novelty compared to this work.

3.      Graphical abstract is not very clear.

4.      Figure arrangement in Figures 5, 6, and 7 is totally unacceptable. The author needs to revise this section and make it eye-catching. Some of them could be presented separately even though they contain a common point.

5.      I appreciate the author providing a lot of experimental outputs, however, I doubt the figure presentation will limit its attractiveness. Too much space was used and I observe large free space in between figures. I strongly recommend using the area wisely. Not only the writing section but also the way of presentation has credit from readers. Need revision for all figures.

6.      I suggest making Figure 12 into two parts.

7.      Figure caption should be enriched with sufficient information.

8.      Overall, I have a good feeling about this work. Too much investigation and analysis were performed in this work.

The manuscript is well-organized with sufficient information in it. Mind that, the language has to be improved with regards to the grammar or extremely long sentences and flow of discussion.

Round 2

Reviewer 1 Report

The authors have carefully addressed all of my previous critiques, revised the manuscript accordingly, and now present a more convincing case for publication. The rather weak initial evidence (at least to me) that biotin-linked Ru particles are more specific for CRC cells than normal counterparts is now presented more convincingly. Also, some steps have been taken to improve the presentation of the vast amounts of accumulated data. Thus, these biotin-linked Ru particles appear to be rather promising for anti-CRC therapy. 

Reviewer 2 Report

The authors addressed all issues.

Minor editing of English language required